# Regulation of Oncogenic Targets by the Tumor-Suppressive *miR-139* Duplex (*miR-139-5p* and *miR-139-3p*) in Renal Cell Carcinoma

**DOI:** 10.3390/biomedicines8120599

**Published:** 2020-12-12

**Authors:** Reona Okada, Yusuke Goto, Yasutaka Yamada, Mayuko Kato, Shunichi Asai, Shogo Moriya, Tomohiko Ichikawa, Naohiko Seki

**Affiliations:** 1Department of Functional Genomics, Chiba University Graduate School of Medicine, Chiba 260-8670, Japan; reonaokada@chiba-u.jp (R.O.); caxa1117@chiba-u.jp (Y.G.); yasutaka_yamada@dfci.harvard.edu (Y.Y.); mayukokato@chiba-u.jp (M.K.); cada5015@chiba-u.jp (S.A.); 2Department of Urology, Chiba University Graduate School of Medicine, Chiba 260-8670, Japan; tomohiko_ichikawa@faculty.chiba-u.jp; 3Department of Biochemistry and Genetics, Chiba University Graduate School of Medicine, Chiba 260-8670, Japan; moriya.shogo@chiba-u.jp

**Keywords:** *miR-139-5p*, *miR-139-3p*, renal cell carcinoma (RCC), microRNA, tumor suppressor, paxillin (*PXN*)

## Abstract

We previously found that both the guide and passenger strands of the *miR-139* duplex (*miR-139-5p* and *miR-139-3p*, respectively) were downregulated in cancer tissues. Analysis of TCGA datasets revealed that low expression of *miR-139-5p* (*p* < 0.0001) and *miR-139-3p* (*p* < 0.0001) was closely associated with 5-year survival rates of patients with renal cell carcinoma (RCC). Ectopic expression assays showed that *miR-139-5p* and *miR-139-3p* acted as tumor-suppressive miRNAs in RCC cells. Here, 19 and 22 genes were identified as putative targets of *miR-139-5p* and *miR-139-3p* in RCC cells, respectively. Among these genes, high expression of *PLXDC1*, *TET3*, *PXN*, *ARHGEF19*, *ELK1*, *DCBLD1*, *IKBKB*, and *CSF1* significantly predicted shorter survival in RCC patients according to TCGA analyses (*p* < 0.05). Importantly, the expression levels of four of these genes, *PXN*, *ARHGEF19*, *ELK1*, and *IKBKB*, were independent prognostic factors for patient survival (*p* < 0.05). We focused on *PXN* (paxillin) and investigated its potential oncogenic role in RCC cells. *PXN* knockdown significantly inhibited cancer cell migration and invasion, possibly by regulating epithelial–mesenchymal transition. Involvement of the *miR-139-3p* passenger strand in RCC molecular pathogenesis is a new concept. Analyses of tumor-suppressive-miRNA-mediated molecular networks provide important insights into the molecular pathogenesis of RCC.

## 1. Introduction

Renal cell carcinoma (RCC) accounts for approximately 3% of adult malignancies, with more than 330,000 cases newly diagnosed in 2018 and more than 100,000 deaths annually [1]. Clear cell RCC is the most common histological subtype of RCC, accounting for approximately 75% of all cases [2]. Approximately 20–30% of RCC patients have metastatic lesions at diagnosis, and thus the 5-year survival rate of these patients is less than 20% [2,3]. In addition, more than 20% of patients develop metastases during the postoperative follow-up period [4]. Elucidation of the molecular mechanisms involved in distant metastasis of RCC will contribute to the development of new diagnostic and therapeutic strategies.

MicroRNAs (miRNAs) are classified as noncoding RNAs approximately 18–25 bases long that are found in a wide range of organisms, from plants to humans [5]. miRNAs bind to the 3′-UTR of their target genes (protein coding and noncoding) in a sequence-dependent manner to regulate their expression. miRNAs are involved in various intracellular processes [6,7]. A unique property of these molecules is that a single miRNA can control a vast number of genes in normal and diseased cells [6,7]. Therefore, aberrant expression of miRNAs disrupts intracellular transcriptional networks, which in turn causes human diseases. In cancer cells, aberrant expression of miRNAs is closely associated with cancer cell progression, metastasis, and drug resistance [8,9].

According to the original theory regarding miRNA biogenesis, the passenger strand of the miRNA duplex is degraded and therefore considered to have no function [6,7]. Our RNA-sequencing-based signatures refute this concept, and our recent studies showed downregulation of both the guide and passenger strands of miRNA duplexes (e.g., *miR-30c*, *miR-99a*, *miR-101*, *miR-143*, *miR-145*, and *miR-150*) in cancer tissues [10,11,12]. Furthermore, ectopic expression assays revealed that the passenger strand of some miRNAs, similar to the guide strand, functions as tumor-suppressive miRNAs, regulating many oncogenes intracellularly [13,14,15,16]. Involvement of miRNA passenger strands in the molecular pathogenesis of human cancers is a new concept, and additional research on passenger strands is needed. A recent study confirmed that both strands of miRNAs are functional and, despite having different seed sequences, cooperate to control molecular pathways across cancer types [17].

In this study, we focused on *miR-139-5p* (the guide strand) and *miR-139-3p* (the passenger strand) and investigated their functional significance. We searched for oncogenes that are controlled by *miR-139-3p* in RCC cells in public databases and identified 22 genes as putative targets of *miR-139-3p* regulation. Among these genes, high expression of *PXN*, *ELK1*, *ARHGEF19*, *DCBLD1*, *IKBKB*, and *CSF1* significantly predicted shorter survival in RCC patients according to The Cancer Genome Atlas (TCGA) analyses (*p* < 0.05). We discuss the oncogenic role of PXN in RCC cells.

## 2. Materials and Methods

### 2.1. Human RCC Cell Lines

The A498 and 786-O human RCC cell lines were used in this study and were obtained from the Japanese Collection of Research Bioresources Cell Bank. Cell maintenance was performed as we described previously [15].

### 2.2. RNA Extraction and Quantitative Reverse Transcription-Polymerase Chain Reaction

RNA was extracted from cell lines using TRIzol reagent (Invitrogen, Carlsbad, CA, USA) according to the manufacturer’s protocol. For gene expression assays, reverse transcription of the RNA was performed using the High-Capacity cDNA Reverse Transcription Kit (Applied Biosystems, Waltham, MA, USA) and TaqMan Gene Expression Assays (Applied Biosystems), according to our previous studies [13,14,15,16]. The expression of *GAPDH* was evaluated as the internal control. We used the CFX Connect Real-Time PCR Detection System (Bio-Rad, Hercules, CA, USA). The TaqMan primers and probes used in this study are listed in Appendix A.

### 2.3. Transfection of miRNAs, siRNAs, and Plasmid Vectors into RCC cells

Transfection of miRNAs and siRNAs into RCC cells was performed using Lipofectamine RNAiMAX reagent (Invitrogen) according to our previous studies [13,14,15,16]. miRNAs or siRNAs were added to a final concentration of 10 nM. Plasmid vectors were transfected into the cells using Lipofectamine 2000 (Invitrogen). The final concentration was 50 ng/well.

### 2.4. Functional Assays (Cell Proliferation, Migration, and Invasion Assays) in RCC cells

XTT assay for cell proliferation, wound healing assay for migration, and Matrigel chamber assay for invasion were performed using RCC cells as described previously [13,14,15,16]. The reagents used are listed in Appendix A.

### 2.5. Identification of miR-139-5p and miR-139-3p Gene Targets in RCC Cells

The search strategy used to identify miRNA targets is summarized in Appendix A. The expression profiles of *miR-139-5p*/*miR-139-3p*-transfected A498 and 786-O cells from the Gene Expression Omnibus (GEO) GSE129043 dataset were used. GSE36895 datasets were also utilized (details below). The TargetScanHuman database (http://www.targetscan.org/vert_72/) was used to predict miRNA-binding sites.

### 2.6. In silico Analysis of RCC Public Databases

For comparison of gene expression levels between normal and cancer tissues, we utilized the GSE36895 datasets obtained from GEO. GSE36895 contains mRNA microarray data from clear cell RCC tumors, normal kidney tissues, and mouse tumor graft samples obtained using Affymetrix U133 Plus 2.0 whole-genome chips (Affymetrix, Santa Clara, CA, USA). Expression is shown as signal intensities, and for each gene with multiple probes, the mean intensity value was used. In addition, RNA sequencing data from TCGA—Kidney renal clear cell carcinoma (TCGA-KIRC) RNA sequencing datasets were utilized to re-evaluate gene expression levels in normal versus tumor samples.

### 2.7. Clinicopathological Analysis of RCC

For Kaplan–Meier analyses of overall survival, we downloaded TCGA clinical data (Firehose Legacy) from cBioportal (https://www.cbioportal.org). The patients were divided into two groups by median expression for each gene, according to the data collected from OncoLnc (http://www.oncolnc.org). R version 4.0.2 (R Foundation for Statistical Computing, Vienna, Austria) was used for the analyses.

Multivariate Cox regression analyses were also performed using TCGA-KIRC clinical data and survival data according to the expression level of each gene from OncoLnc to identify factors associated with RCC patient survival. In addition to gene expression, the tumor stage, pathological grade, and age group were evaluated as potential independent prognostic factors. The multivariate analyses were performed using JMP Pro 15.0.0 (SAS Institute Inc., Cary, NC, USA).

We performed gene set enrichment analysis (http://software.broadinstitute.org/gsea/index.jsp) to obtain lists of differentially expressed genes between high and low *PXN* expression groups in the TCGA-KIRC cohort.

### 2.8. Plasmid Construction and Dual-Luciferase Reporter Assays

psiCHECK-2 plasmid vectors (Promega, Madison, WI, USA) harboring the wild-type or a deletion sequence of the *miR-139-3p*-binding site within the 3′-UTR of *PXN* were prepared. The predicted binding site sequence was obtained from the TargetScanHuman database (release 7.2). Cells were co-transfected with *miR-139-3p* and the plasmid vectors for 36 h, after which firefly and *Renilla* luciferase activities in cell lysates were measured consecutively using the Dual-Luciferase Reporter Assay System (Promega). *Renilla* luciferase activities are expressed as normalized values to firefly luciferase activities. The dual-luciferase reporter assay procedure was described in our previous studies [13,14,15,16].

### 2.9. Western Blotting

Cell lysates were prepared 48 h after transfection with RIPA buffer (Nacalai Tesque, Chukyo-ku, Kyoto, Japan). Then, 20 μg of protein lysates were separated on 4–15% Mini-PROTEAN TGX Precast Gels (Bio-Rad), and transferred to Immun-Blot PVDF membranes (Bio-Rad). Blocking was performed with Blocking One (Nacalai Tesque) for 30 min. The antibodies used in this study are shown in Appendix A.

### 2.10. Statistical Analyses

For comparisons among multiple groups, Dunnett’s test was applied. The statistical analyses were performed using JMP Pro 15. Significance levels were set to *p* < 0.05 if not otherwise mentioned.

## 3. Results

### 3.1. Analysis of miR-139-5p and miR-139-3p Expression Levels in Clinical RCC Tissues and Their Clinical Significance

The expression levels of *miR-139-5p* and *miR-139-3p* were evaluated in RNA sequencing data from RCC tissue samples obtained from TCGA. *miR-139-5p* and *miR-139-3p* were significantly downregulated in RCC tissues compared with normal tissues (*p* < 0.0001 and *p* < 0.0001, respectively; Figure 1A). Cohort analyses using the TCGA-KIRC datasets revealed that low expression of *miR-139-5p* and *miR-139-3p* was associated with poorer survival in patients with RCC (*p* < 0.0001 and *p* < 0.0001, respectively; Figure 1B).

### 3.2. Tumor-Suppressive Functions of miR-139-5p and miR-139-3p in RCC Cells

To investigate the tumor-suppressive functions of *miR-139-5p* and *miR-139-3p* in RCC cells, we assessed cell proliferation, migration, and invasion after ectopic transfection of *miR-139-5p* and *miR-139-3p* into A498 and 786-O cells. Ectopic expression of the two miRNAs did not significantly affect the proliferation of RCC cells (Figure 2A). In contrast, the expression of these miRNAs significantly inhibited the migration and invasive abilities of RCC cells (Figure 2B,C).

### 3.3. Identification of Putative Oncogenic Targets Regulated by miR-139-3p and miR-139-3p in RCC Cells

To identify the genes regulated by *miR-139-5p* and *miR-139-3p* in RCC cells, we integrated three datasets. First, we obtained RNA microarray data from *miR-139-5p-* or *miR-139-3p*-transfected A498 or 786-O cells (GSE129043). Second, we used data from the TargetScanHuman database (release 7.2) to obtain annotated putative targets regulated by each miRNA strand. Third, we extracted genes highly expressed in clinical specimens of RCC from GSE36895.

A schematic of the strategy used to narrow down the gene list is shown in Appendix A. A total of 19 genes regulated by *miR-139-5p* and 22 by *miR-139-3p* were finally selected (Table 1A,B).

### 3.4. Clinical Significance of miR-139 Target Genes in RCC Pathogenesis

Kaplan–Meier analyses of 5-year overall survival were performed according to high versus low expression of *miR-139* target genes. The high expression of two *miR-139-5p* (*PLXDC1* and *TET3*) and six *miR-139-3p* target genes (*PXN*, *ARHGEF19*, *ELK1*, *CSF1*, *IKBKB*, and *DCBLD1*) was found to be related to better 5-year overall survival rates of the patients (*p* < 0.05, Table 1A,B, and Figure 3).

Furthermore, multivariate analyses identified four genes (*PXN*, *ARHGEF19*, *ELK1*, and *IKBKB*) as independent prognostic factors for patient survival (*p* < 0.05; Figure 4). We validated the expression levels of these target genes using TCGA-KIRC RNA sequencing dataset. All target genes except *ARHGEF19* were upregulated in cancer tissues in this cohort (Figure 5).

For subsequent analyses, we focused on miR-139-3p (the passenger strand) target genes, according to our previous studies emphasizing the novel roles of miRNA passenger strands. Among these target genes, we focused on *PXN*, which showed the most significant association with RCC patient survival in the multivariate analysis and has been reported previously as an oncogene in other types of cancers [18,19].

### 3.5. Direct Regulation of PXN by miR-139-3p in RCC Cells

In RCC cells transfected with *miR-139-3p*, both PXN mRNA and protein levels were significantly downregulated (Figure 6A,B). Western blot images are shown in Appendix A. To validate that *miR-139-3p* binds directly to *PXN* to downregulate its expression, we performed dual-luciferase reporter assays. A498 and 786-O cells were co-transfected with plasmid vectors and *miR-139-3p*. We used two different plasmid vectors: one containing the partial wild-type sequence of the *miR-139-3p*-binding site predicted by TargetScanHuman database (“wild-type sequence” in Figure 6C) and the other containing this sequence lacking the binding site (“deletion-type sequence” in Figure 6C). Luciferase activity was significantly reduced in cells transfected with the wild-type sequence but not in cells transfected with the deletion sequence (Figure 6D). These results suggest that *miR-139-3p* directly binds to the 3’-UTR of *PXN*.

### 3.6. PXN Knockdown Assays in RCC Cells

We assessed the oncogenic functions of PXN in RCC cells by performing knockdown using two small interfering RNAs (siRNAs) targeting *PXN*. The mRNA and protein levels of PXN were successfully downregulated by either siRNAs in A498 and 786-O cells (Figure 7A,B). Western blot images are shown in Appendix A.

Next, functional assays were performed in RCC cells transfected with these siRNAs. Similar to *miR-139-3p* transfection, transfection of the siRNAs did not suppress cell proliferation (Figure 7C). However, cell migration and invasion were significantly suppressed by siRNA transfection in both cell lines (Figure 7D,E).

### 3.7. PXN-Mediated Pathways in RCC Cells

We performed gene set enrichment analysis to determine differentially expressed genes between the high and low *PXN* expression groups of the TCGA-KIRC cohort. In support of the functional assays using siRNAs, the most enriched signaling pathway in the high *PXN* expression group was “epithelial–mesenchymal transition” (Figure 8 and Appendix A). Other significantly enriched pathways (FDR *q*-value < 0.05) were “hypoxia”, “KRAS signaling”, “myogenesis”, “angiogenesis”, and “apical junction”, supporting the hypothesis that *PXN* is related to the metastatic ability of cancer cells (Figure 7 and Appendix A).

## 4. Discussion

Recent RNA sequencing techniques have accelerated the establishment of miRNA expression signatures. Our recent studies revealed that some passenger strands of miRNAs (e.g., *miR-30c-2-3p*, *miR-99a-3p*, *miR-101-5p*, *miR-143-5p*, *miR-145-3p*, and *miR-150-3p*) act as tumor-suppressive miRNAs by targeting several oncogenes in a wide range of cancers [10,11,12,13,14,15,16]. Involvement of the passenger strands of miRNAs in cancer pathogenesis is a new development in cancer research.

A recent in silico study (analysis of molecular profiles in more than 5200 patient samples from 14 different cancers) revealed that both strands of miRNAs are functional across different cancer types. For example, downregulation of both strands of *miR-30a* and *miR-145* was frequently observed in multiple cancers, and their downregulation enhanced aberrant expression of cell cycle-related genes. These malignant events were found to affect the prognosis of cancer patients [17]. Simultaneous analysis of both strands of miRNA duplexes will lead to elucidation of a new molecular mechanism in cancer cells.

In this study, we showed that both strands of the *miR-139* duplex acted as tumor-suppressive miRNAs in RCC cells. Accumulating evidence has shown a tumor-suppressive role of *miR-139-5p* in multiple types of cancers, including RCC [20,21,22]. Notably, *miR-139-5p* target genes are involved in cancer activation pathways, e.g., the PI3K/AKT/mTORC1, Wnt/β-catenin, and RTK/RAS/MAPK pathways [23,24,25,26,27]. Therefore, *miR-139-5p* plays a central role in controlling the malignant transformation of human cancers.

In contrast to *miR-139-5p*, few studies have evaluated the importance of *miR-139-3p* in cancer because it is a passenger strand. Recently, it was reported that *miR-139-3p* directly regulates *KDM5B* (lysine demethylase 5B), a key regulator of histone 3 lysine 4 demethylation in laryngeal squamous cell carcinoma [28]. Ectopic expression of *miR-139-3p* suppressed cancer cell malignant behavior by inhibiting the Wnt/β-catenin pathway [28]. In the HeLa cervical cancer cell line, *miR-139-3p* was reduced upon increasing expression of circular RNA hsa_circ_0031288 [29]. In turn, the decreased expression of *miR-139-3p* resulted in increased expression of B-cell CLL/lymphoma 6, as a target molecule. These regulatory effects promoted malignant transformation of HeLa cells. In bladder cells, both strands of the *miR-139* duplex were downregulated in cancer tissues and exhibited tumor-suppressive effects [30]. Interestingly, matrix metalloprotease 11 (*MMP11*) was directly regulated by both *miR-139* strands, and knockdown of *MMP11* attenuated the aggressive phenotype of bladder cancer cells [30]. Our study is the first to show that *miR-139-3p* (the passenger strand) is involved in RCC pathogenesis, and both strands of the *miR-139* duplex are pivotal players in RCC oncogenesis.

We are also interested in the presence of oncogenes controlled by tumor-suppressive *miR-139-5p* and *miR-139-3p* in RCC cells. A total of 19 and 22 genes were identified as putative targets of *miR-139-5p* and *miR-139-3p* regulation in RCC cells, respectively. Taking advantage of survival data from TCGA datasets, we performed multivariate analysis and found that, among the *miR-139* duplex target genes, *PXN*, *ELK1*, *ARHGEF19*, and *IKBKB* were independent prognostic factors for RCC patient survival. Further genomic analyses of these genes will contribute to elucidating the molecular pathogenesis of RCC.

ELK1 is a well-established transcription factor that is phosphorylated by MAPKs and induces transcription of the c-fos proto-oncogene [31,32]. *ARHGEF19* is a RhoGEF that reportedly activates the MAPK pathway and interacts with BRAF in lung cancer [33]. The role of *ARHGEF19* in RCC is not well characterized, but the interaction of *ARHGEF19* with *ELK1* via the MAPK pathway may play a role in RCC development. *IKBKB* encodes IKK-beta, which phosphorylates the inhibitor of NF-kB, resulting in activation of the NF-kB pathway [34,35]. NF-kB and its downstream inflammatory signaling pathway are closely related to RCC carcinogenesis and aggressiveness [36].

In this study, we focused on *PXN* (paxillin) because its expression was found to be directly controlled by *miR-139-3p* in RCC cells and strongly related to RCC molecular pathogenesis. PXN is a focal adhesion scaffold/adaptor protein that contains five LD domains (leucine-aspartate motifs) located at the N-terminus and four cysteine–histidine-rich LIM domains at the C-terminus [37,38]. The LD domains act as a docking site for focal adhesion-related proteins, e.g., Src (tyrosine-protein kinase), FAK (focal adhesion kinase), PAK (p21-activated kinase), and ILK (integrin-linked kinase) [37,38]. The LIM domains act as binding sites for protein–protein interactions [37,38]. The pathways activated by PXN enhance cancer cell malignant progression and metastasis in a wide range of cancers [39]. Moreover, two key proteins in focal adhesion complexes, PXN and integrin B4, directly bind to each other, and this complex enhanced cisplatin resistance in lung cancer cells [40]. Another study showed that PXN phosphorylation may contribute to cisplatin resistance via the ERK-mediated activation of Bcl-2 transcription in lung cancer [41]. PXN-mediated pathways may be therapeutic targets for attenuating drug-resistance in cancer cells.

In our GSEA analysis, *PXN*-high-expressed RCC specimen was enriched with epithelial-mesenchymal transition (EMT) signaling pathways. A previous report has shown the suppressive effect of *PXN* on EMT pathway in cell lines [42]. These findings suggest the speculation that inhibitions of cancer cell migration and invasion by *PXN* knockdown in our study are due to the regulation of epithelial-mesenchymal transition.

It has been reported that several miRNAs (e.g., *miR-132*, *miR-137*, *miR-145*, *miR-212*, and *miR-218*) directly control *PXN* expression in cancer cells [43,44,45,46,47]. Among these miRNAs, we previously reported the downregulation of *miR-145* and *miR-218* in RCC cells [48,49]. More recently, *DLX6-AS1* (a long noncoding RNA that adsorbs *miRNA-199b*) promoted epithelial–mesenchymal transition and cisplatin resistance via the *miR-199b-5p*/*PXN* axis in breast cancer cells [50]. This is the first report that *PXN* is directly regulated by tumor-suppressive *miR-139-3p* in RCC cells. Non-coding RNA-mediated epigenetic regulation of *PXN* expression will be assessed in the future.

## 5. Conclusions

Both strands of the *miR-139* duplex are closely involved in RCC oncogenesis. This is the first report to reveal that *miR-139-3p* (the passenger strand) acts as a tumor-suppressive miRNA in RCC. Several genes are controlled by *miR-139* and contribute to RCC molecular pathogenesis. Notably, the expression of PXN was directly regulated by *miR-139-3p*, and its overexpression enhanced RCC malignant transformation. Analyses of tumor-suppressive miRNAs (including the passenger strands) will contribute to the elucidation of new molecular networks in RCC.

## Figures and Tables

**Figure 1 biomedicines-08-00599-f001:**
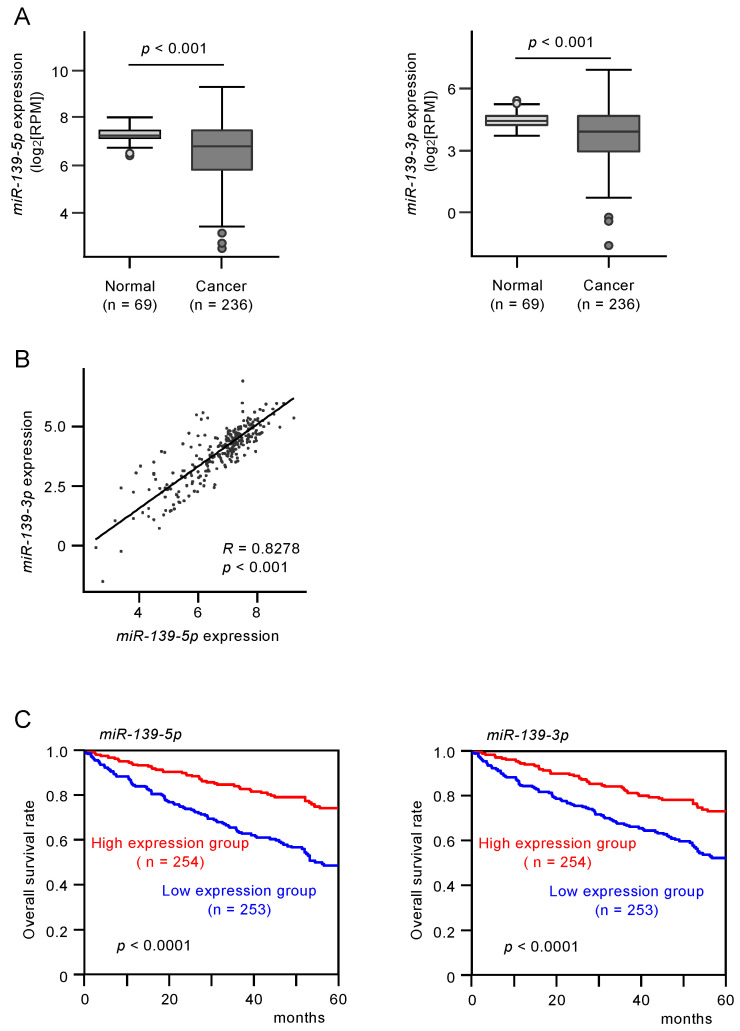
The expression and clinical significance of *miR-139-5p* and *miR-139-3p* in renal cell carcinoma (RCC) clinical specimens. (**A**) Expression of *miR-139-5p* and *miR-139-3p* were significantly reduced in The Cancer Genome Atlas—Kidney Renal Clear Cell Carcinoma (TCGA-KIRC) cancer specimens compared with adjacent normal specimens (*p* < 0.001). (**B**) Spearman’s rank test showed positive correlations between the expression levels of *miR-139-5p* and *miR-139-3p* in TCGA-KIRC clinical specimens (R = 0.8278, *p* < 0.001). (**C**) Kaplan–Meier survival curves of patients from TCGA-KIRC cohort. Patients were divided into two groups according to the median expression levels of miR-139-5p or miR-139-3p: high- and low-expression groups. Both *miR-139-5p* and *miR-139-3p* expression levels were significantly associated with the 5-year survival rate of RCC patients (*p* < 0.0001).

**Figure 2 biomedicines-08-00599-f002:**
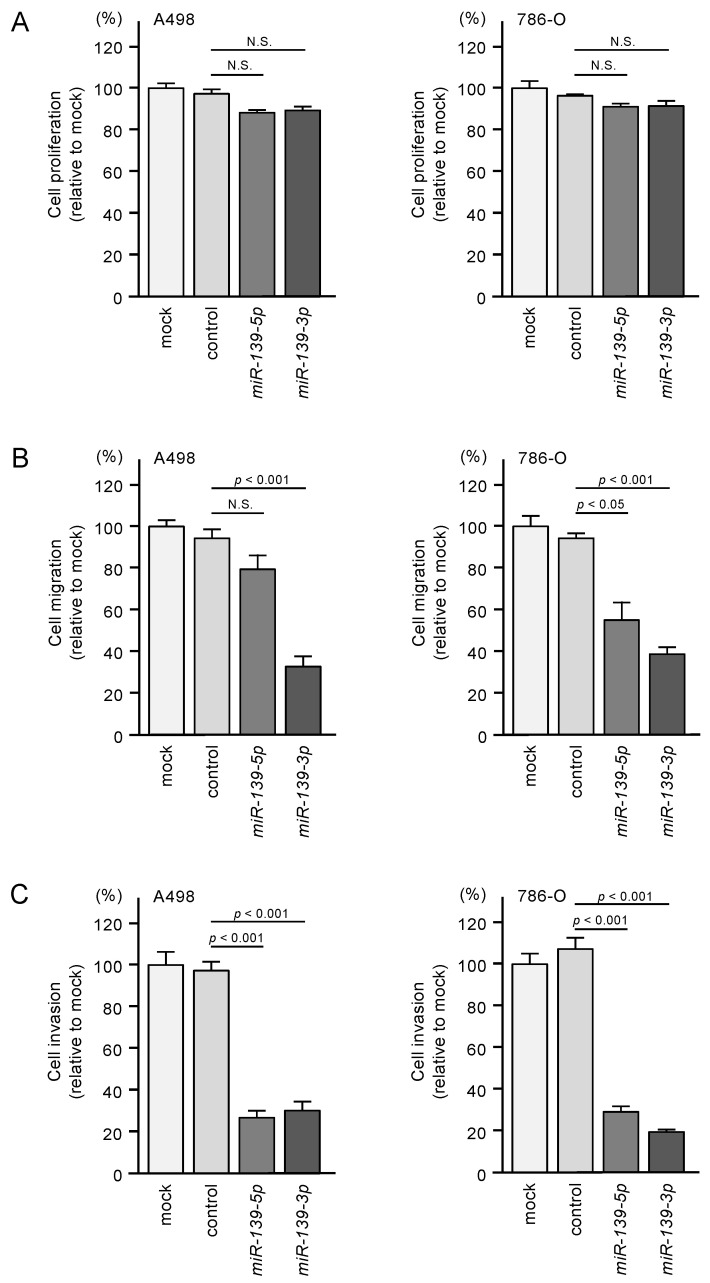
Functional assays of cell proliferation, migration, and invasion following ectopic expression of *miR-139-5p* and *miR-139-3p* in RCC cell lines A498 and 786-O. N.S.: not significant. (**A**) Cell proliferation was assessed using XTT assays. Data were collected 72 h after miRNA transfection. (**B**) Cell migration was assessed using wound healing assays. (**C**) Cell invasions were determined 48 h after seeding miRNA-transfected cells into chambers using Matrigel invasion assays.

**Figure 3 biomedicines-08-00599-f003:**
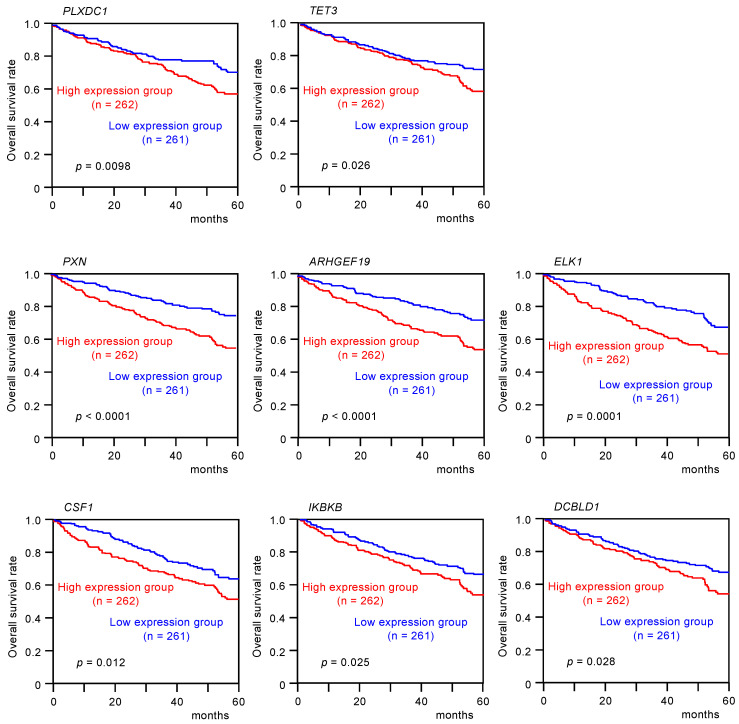
Clinical significance of *miR-139-5p* or *miR-139-3p* target genes in TCGA-KIRC database. High expression of two *miR-139-5p* target genes (*PLXDC1* and *TET3*) and six *miR-139-3p* target genes (*PXN*, *ARHGEF19*, *ELK1*, *CSF1*, *IKBKB*, and *DCBLD1*) were significantly associated with poor prognosis in patients with RCC. Kaplan–Meier curves for 5-year overall survival according to the expression of each *miR-139-3p* target gene are shown.

**Figure 4 biomedicines-08-00599-f004:**
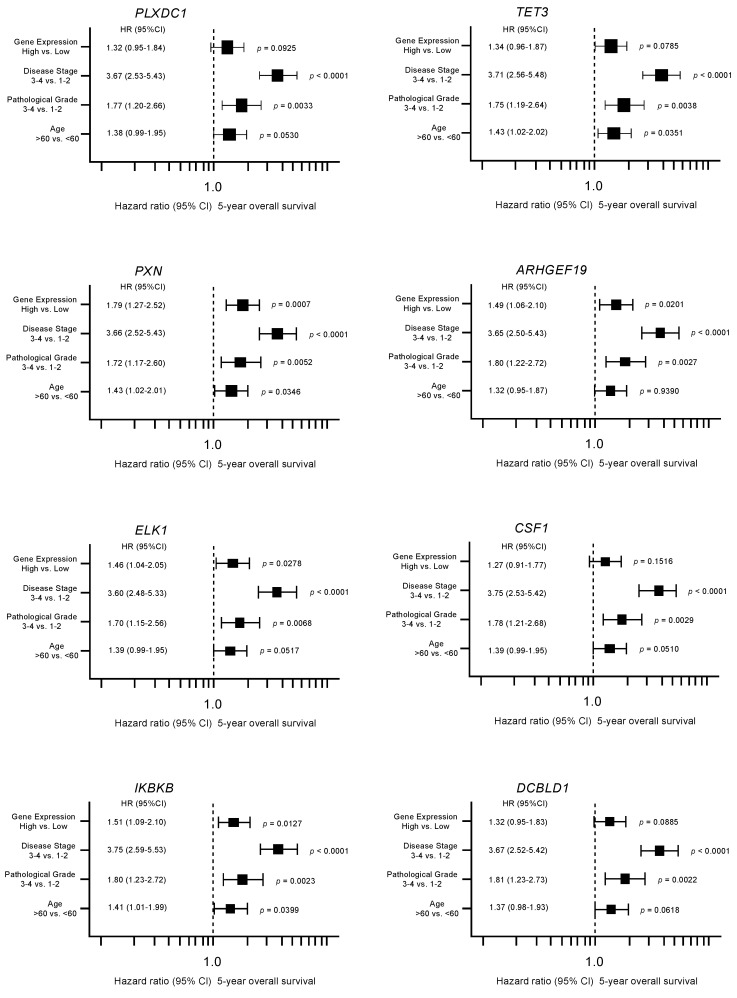
Forest plot showing the results of multivariate analyses of eight genes (*PLXDC1*, *TET3, PXN*, *ARHGEF19*, *ELK1*, *CSF1*, *IKBKB*, and *DCBLD1*). In addition to the gene expression level, the tumor stage, pathological grade, and age group were evaluated as potential independent factors associated with survival. Numbers of cases per each group are shown in Appendix A.

**Figure 5 biomedicines-08-00599-f005:**
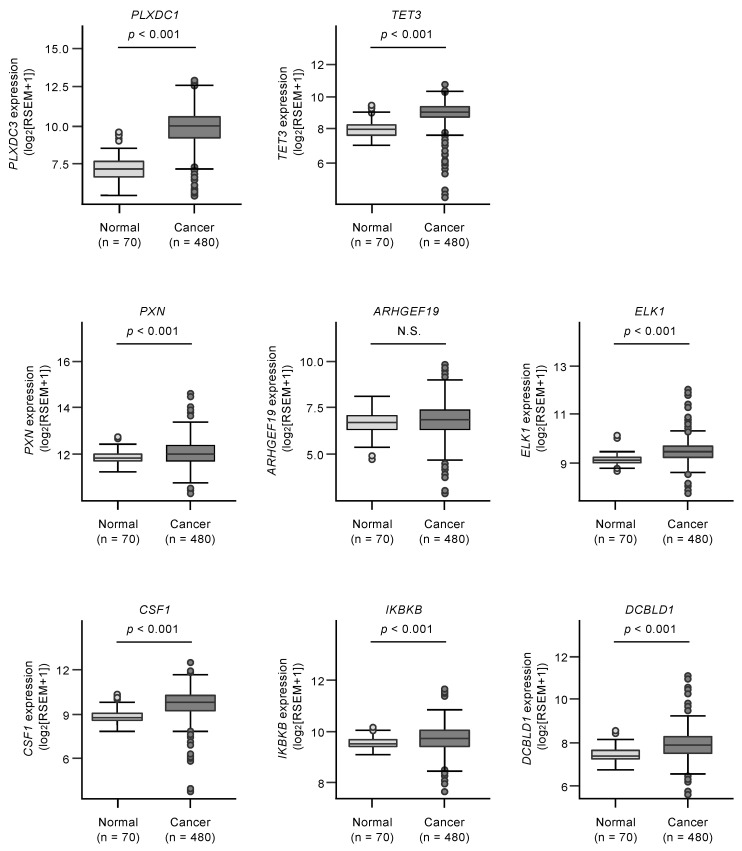
Expression levels of *miR-139-5p* or *miR-139-3p* target genes (*PLXDC1* and *TET3* for *miR-139-5p*, *PXN*, *ARHGEF19*, *ELK1*, *CSF1*, *IKBKB*, and *DCBLD1* for *miR-139-3p*) in TCGA-KIRC cohort. Seven genes (*PLXDC1*, *TET3*, *PXN*, *ELK1*, *CSF1*, *IKBKB*, and *DCBLD1*) were confirmed to be significantly upregulated in clinical specimens. N.S.: not significant.

**Figure 6 biomedicines-08-00599-f006:**
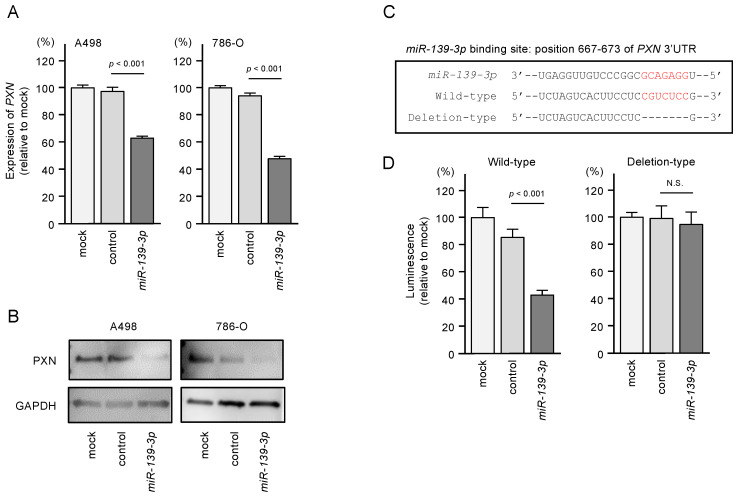
Expression of paxillin (PXN) was regulated directly by *miR-139-3p* in RCC cells. (**A**) Expression of *PXN* mRNA was significantly suppressed in *miR-139-3p*-transfected A498 and 786-O cells (48 h after transfection). Expression of *GAPDH* was used as an internal control. (**B**) Expression of PXN protein was reduced in *miR-139-3p-*transfected RCC cells (48 h after transfection). GAPDH was used as a loading control. (**C**) The TargetScanHuman database predicted one putative *miR-139-3p*-binding site in the 3’-UTR of *PXN*. (**D**) Dual-luciferase reporter assays showed decreased luminescence activity in RCC cells co-transfected with *miR-139-3p* together with a vector harboring the “wild-type sequence”. Normalized data were calculated as *Renilla*/firefly luciferase activity ratios. N.S.: not significant.

**Figure 7 biomedicines-08-00599-f007:**
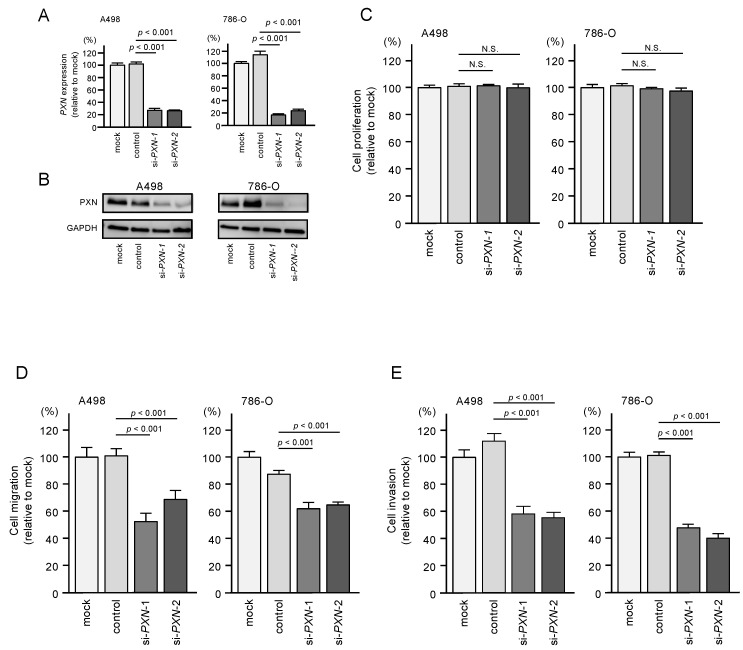
Effects of *PXN* knockdown on cell proliferation, migration, and invasion in RCC cells. (**A**,**B**) Expression of PXN was successfully reduced after siRNA transfection in RCC cells. (**C**) Cell proliferation was determined by XTT assays. N.S.: not significant. (**D**) Cell migration was determined by wound healing assays. (**E**) Cell invasion was determined by Matrigel invasion assays.

**Figure 8 biomedicines-08-00599-f008:**
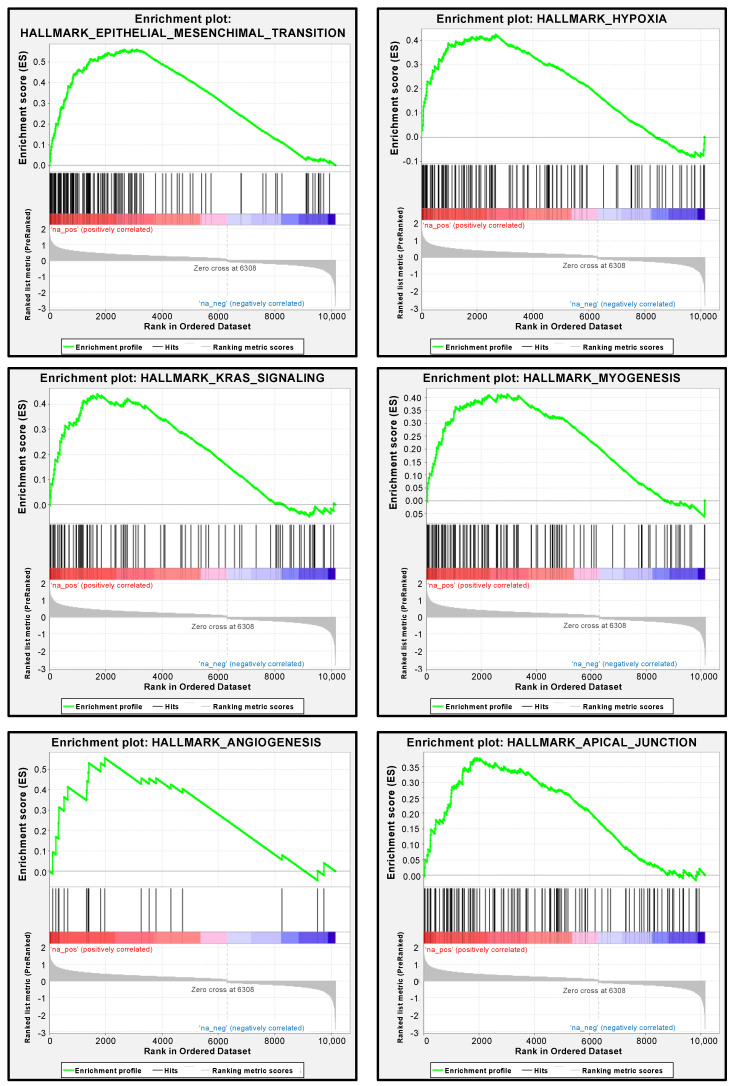
Pathways enriched among the differentially expressed genes in the high *PXN* expression group according to gene set enrichment analysis. The six significantly enriched pathways (FDR *q*-value < 0.05) are shown.

**Table 1 biomedicines-08-00599-t001:** **A.** Candidate target genes of *miR-139-5p*. **B.** Candidate target genes of *miR-139-3p.*

**A.** Candidate target genes of *miR-139-5p*							
**Gene** **Symbol**	**Gene Name**	**Entrez** **Gene** **ID**	***miR-139-5p*** **-** **Transfected** **A498 Cells** **(log2 FC)**	***miR-139-5p*** **-** **Transfected** **786-O Cells** **(log2 FC)**	**GSE36895** **(log2 FC)**	**Total** **Binding** **Sites**	**TCGA** **5y OS** ***p*-Value**
*PLXDC1*	plexin domain containing 1	57125	−0.615791	−1.455703	1.6528159	1	0.0098
*TET3*	tet methylcytosine dioxygenase 3	200424	−1.074818	−1.152835	0.7350367	1	0.0261
*IRF4*	interferon regulatory factor 4	3662	−2.491284	−1.563216	0.72956836	1	0.0575
*RAB27B*	RAB27B, member RAS oncogene family	5874	−0.620716	−0.970745	0.7823266	1	0.1244
*FCHSD2*	FCH and double SH3 domains 2	9873	−1.834753	−1.364281	0.5673638	1	0.1565
*DMD*	dystrophin	1756	−0.98734	−1.12498	0.46094477	1	0.3891
*APOL6*	apolipoprotein L, 6	80830	−1.363298	−0.574163	0.43652818	2	0.4733
*AP1S2*	adaptor-related protein complex 1, sigma 2 subunit	8905	−0.571341	−0.63469	0.57979524	1	0.5072
*PTPRU*	protein tyrosine phosphatase, receptor type, U	10076	−1.430874	−0.883479	0.8481535	1	0.6152
*TRAT1*	T cell receptor associated transmembrane adaptor 1	50852	−1.192173	−1.781637	1.9733018	2	0.7395
*SLC39A14*	solute carrier family 39 (zinc transporter), member 14	23516	−0.536152	−0.705082	1.0408258	1	0.8125
*OTUD4*	OTU deubiquitinase 4	54726	−1.720465	−1.59549	0.21931966	1	0.9338
*CDCA7L*	cell division cycle associated 7-like	55536	−1.478013	−0.564377	2.1733913	1	0.969
*ZNF678*	zinc finger protein 678	339500	−1.585917	−0.621832	0.25128952	2	0.0086 *
*FGFBP2*	fibroblast growth factor binding protein 2	83888	−0.830757	−1.317887	1.5742466	1	0.0058 *
*ATP2B2*	ATPase, Ca++ transporting, plasma membrane 2	491	−1.109889	−2.668194	1.3068246	3	0.0050 *
*EML1*	echinoderm microtubule associated protein like 1	2009	−0.56874	−0.541978	0.38891175	2	0.0014 *
*PCSK5*	proprotein convertase subtilisin/kexin type 5	5125	−1.751102	−0.577052	0.57546955	1	0.0006 *
*FAM168A*	family with sequence similarity 168, member A	23201	−1.390649	−0.959992	0.25754136	1	0.0002 *
**B.** Candidate target genes of *miR-139-3p*							
**Gene** **Symbol**	**Gene Name**	**Entrez** **Gene** **ID**	***miR-139-3p*** **-** **Transfected** **A498 Cells** **(log2 FC)**	***miR-139-3p*** **-** **Transfected** **786-O Cells** **(log2 FC)**	**GSE36895** **(log2 FC)**	**Total** **Binding** **Sites**	**TCGA** **5y OS** ***p*-Value**
*PXN*	paxillin	5829	−1.164181	−0.707167	0.481819	2	<0.0001
*ARHGEF19*	Rho guanine nucleotide exchange factor (GEF) 19	128272	−0.607603	−1.826996	1.1049173	1	<0.0001
*ELK1*	ELK1, member of ETS oncogene family	2002	−1.518329	−0.839984	0.5987212	2	0.0001
*CSF1*	colony stimulating factor 1 (macrophage)	1435	−0.952795	−0.537074	1.0153022	1	0.0124
*IKBKB*	inhibitor of kappa light polypeptide gene enhancer in B-cells, kinase beta	3551	−0.511766	−1.882033	0.23441868	1	0.0251
*DCBLD1*	discoidin, CUB and LCCL domain containing 1	285761	−1.398862	−0.668245	0.28628728	1	0.0285
*SYT11*	synaptotagmin XI	23208	−1.034462	−0.578617	0.4411527	1	0.0556
*SERPINE1*	serpin peptidase inhibitor, clade E (nexin, plasminogen activator inhibitor type 1), member 1	5054	−2.516404	−0.663638	1.8049024	3	0.0731
*KDM6B*	lysine (K)-specific demethylase 6B	23135	−0.62567	−0.733171	0.42303625	1	0.1019
*RASSF5*	Ras association (RalGDS/AF-6) domain family member 5	83593	−1.180982	−4.605049	1.5059676	1	0.3643
*ACBD3*	acyl-CoA binding domain containing 3	64746	−0.688812	−0.542166	0.65537864	1	0.3841
*APOL6*	apolipoprotein L, 6	80830	−1.350307	−0.858142	0.43652818	1	0.4733
*EPN2*	epsin 2	22905	−0.619177	−0.520903	0.29098234	1	0.581
*GIT2*	G protein-coupled receptor kinase interacting ArfGAP 2	9815	−0.64332	−0.544951	1.4113243	1	0.7179
*KIF3C*	kinesin family member 3C	3797	−0.694929	−0.956568	0.29198763	1	0.9148
*ARAP2*	ArfGAP with RhoGAP domain, ankyrin repeat and PH domain 2	116984	−1.161352	−1.258001	0.3943753	1	0.0940 *
*RFX2*	regulatory factor X, 2 (influences HLA class II expression)	5990	−0.937881	−0.580092	1.4766915	1	0.0671 *
*RNF125*	ring finger protein 125, E3 ubiquitin protein ligase	54941	−0.539067	−1.811576	0.6103558	1	0.0394 *
*ARSK*	arylsulfatase family, member K	153642	−1.505581	−0.887476	0.4327301	1	0.0219 *
*STAG2*	stromal antigen 2	10735	−0.527565	−0.563985	0.38703138	1	<0.0001 *
*TNS1*	tensin 1	7145	−0.764289	−0.609893	0.28155625	1	<0.0001 *

* Better prognosis in high-expression group.

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
