# Peer review of "Regulation of Oncogenic Targets by the Tumor-Suppressive miR-139 Duplex (miR-139-5p and miR-139-3p) in Renal Cell Carcinoma"

_biomedicines, 2020, doi:10.3390/biomedicines8120599_

Round 1

Reviewer 1 Report

In this manuscript, the authors have comprehensively analyzed the TCGA  (and other database) and have concluded that, low expression of miR-139-5p and miR-139-3p (p < 0.0001) is closely associated with 5-year survival rates of patients with renal cell carcinoma (RCC). Through in-vitro studies, they have also reported that miR-139-3p (the passenger strand) acts as a tumor-suppressive miRNA in RCC and PXN is directly regulated by miR-139-3p. Overall, the data analyses are very comprehensive, in vitro studies are also well done and support the in silico findings.

However, to further strengthen the statement….PXN knockdown significantly inhibited cancer cell migration, and invasion by “regulating epithelial– mesenchymal transition”,  it would be great if the author could also show the fate of epithelial-mesenchymal transition (EMT) “gene signature” post PXN knockdown.

Few minor comments:

  1. Line 198:… and has been reported previously as an oncogene in other type of cancers….: please cite references.
  2. Line 250:…. expression group was “epithelial–mesenchymal transition” (Figure 8 and Table S3)….: Table S3 is not included in SI. Is it TableS2?

Author Response

Comment 1: ... to further strengthen the statement…. PXN knockdown significantly inhibited cancer cell migration, and invasion by “regulating epithelial– mesenchymal transition”, it would be great if the author could also show the fate of epithelial-mesenchymal transition (EMT) “gene signature” post PXN knockdown.

Response: The reviewer pointed out the critical part of our article. In this article, we used data from clinical specimen and confirmed that PXN-high-expressed RCC specimen was enriched with EMT signaling compared with PXN-low-expressed RCC. As for cell lines, several reports have already indicated the relationship between PXN and EMT pathway, especially, Wen et al. recently showed si-PXN suppressed EMT (ref PMID: 32705241).

According to the reviewer's comment, we changed our article as follows.

PXN knockdown significantly inhibited cancer cell migration and invasion possibly by “regulating epithelial-mesenchymal transition”

Also, we refered to previous reports showing relationship between si-PXN and EMT in the discussion section.

In our GSEA analysis, PXN-high-expressed RCC specimen was enriched with epithelial-mesenchymal transition (EMT) signaling pathways. A previous report has shown the suppressive effect of PXN on EMT pathway in cell lines [42]. These findings suggest the speculation that inhibitions of cancer cell migration and invasion by PXN knockdown in our study are due to regulation of epithelial-mesenchymal transition.

Comment 2: Line 198:… and has been reported previously as an oncogene in other type of cancers….: please cite references.

Response: I appreciate the comment. I added some cite references.

  1. Alpha, K.M., Xu, W., Turner, C.E. Paxillin family of focal adhesion adaptor proteins and regulation of cancer cell invasion. Int Rev Cell Mol Biol 2020, 355, 1-52, doi:10.1016/bs.ircmb.2020.05.003.
  2. Noh, K., Bach, D.H., Choi, H.J., Kim, M.S., Wu, S.Y., Pradeep, S., Ivan, C., Cho, M.S., Bayraktar, E., Rodriguez-Aguayo, C., et al. The hidden role of paxillin: localization to nucleus promotes tumor angiogenesis. Oncogene 2020, doi: 10.1038/s41388-020-01517-3.

Comment 3: Line 250:…. expression group was “epithelial–mesenchymal transition” (Figure 8 and Table S3)….: Table S3 is not included in SI. Is it TableS2?

Response: I am sorry for the mistake. To answer other reviewer’s comment, I added new “Supplemental Table 2”, so I fixed the part as “Supplemental Table 3”

Reviewer 2 Report

This is really comprehensive study.  

The authors combined massive bioinformatic study with experiments to derive the confirmed results. It is worthwhile publishing as it is.

Author Response

Comment: This is really comprehensive study.  

The authors combined massive bioinformatic study with experiments to derive the confirmed results. It is worthwhile publishing as it is.

Response: I appreciate your cooperation for reviewing our article.

Reviewer 3 Report

The molecular role of the passenger strand of the miRNAs is quite a recent finding compared to the guide strands of miRNAs. The miR-139-5p has already been established as a promising biomarker in other cancers. However, the manuscript for the first time describes the molecular role of the passenger strand of the miR-139 as a tumor-suppressive miRNA in RCC. 

The manuscript is well organized, and findings are statistically supported through well designed in vitro experiments. Analysis of molecular networks mediated by tumor-suppressive-miRNA also provides important insights into the molecular pathogenesis of RCC.

Overall, the analyses and results presented in the manuscript will enrich the literature about the role of miRNAs, especially their passenger strands.

I have a few minor comments as following,

  1. Further description of Figure 4 in results would enhance understanding. How many cases/observations (n) were considered per group for the hazard ratio calculations? Supplementary data would add value to support the observations.
  2. It seems there is a typo in the URL, http://www.targetscan.org/ver_72/. Please correct it to http://www.targetscan.org/vert_72/ or as appropriate.
  3. Sheet in "Table S1 reagents.xlsx" file is named as Table S2

Author Response

Comment 1: Further description of Figure 4 in results would enhance understanding. How many cases/observations (n) were considered per group for the hazard ratio calculations? Supplementary data would add value to support the observations.

Response: I have made an additional supplemental table (Supplemental Table 2) to show the number of cases per each group.

In accordance with this, I added a description to the legend of Figure 4 as follows.

Numbers of cases per each group are shown in Supplemental Table 2.

Comment 2: It seems there is a typo in the URL, http://www.targetscan.org/ver_72/. Please correct it to http://www.targetscan.org/vert_72/ or as appropriate.

Response: I am sorry for the mistake. Thank you for pointing out. I fixed the URL.

Comment 3: Sheet in "Table S1 reagents.xlsx" file is named as Table S2

Response: I fixed the sheet name as Table S1.